# Prevalence and Genotyping of *Anaplasma phagocytophilum* Strains from Wild Animals, European Bison (*Bison bonasus*) and Eurasian Moose (*Alces alces*) in Poland

**DOI:** 10.3390/ani12091222

**Published:** 2022-05-09

**Authors:** Anna W. Myczka, Stanisław Kaczor, Katarzyna Filip-Hutsch, Michał Czopowicz, Elwira Plis-Kuprianowicz, Zdzisław Laskowski

**Affiliations:** 1Witold Stefański Institute of Parasitology, Polish Academy of Sciences, Twarda 51/55, 00-818 Warsaw, Poland; laskowz@twarda.pan.pl; 2District Veterinary Inspectorate, Młynarska 45, 38-500 Sanok, Poland; kaczor2007@op.pl; 3Department of Food Hygiene and Public Health Protection, Institute of Veterinary Medicine, Warsaw University of Life Sciences—SGGW, Nowoursynowska 159, 02-776 Warsaw, Poland; katarzyna_filip-hutsch@sggw.edu.pl; 4Division of Veterinary Epidemiology and Economics, Institute of Veterinary Medicine, Warsaw University of Life Sciences—SGGW, Nowoursynowska 159c, 02-776 Warsaw, Poland; mczopowicz@gmail.com; 5Białowieża National Park, Park Pałacowy 11, 17-230 Białowieża, Poland; elwira.plis@bpn.com.pl

**Keywords:** *Anaplasma phagocytophilum*, *16S* rDNA, *groEL*, *ankA*, *Bison bonasus*, *Alces alces*

## Abstract

**Simple Summary:**

The populations of bison and moose, the largest wild ruminants in Poland, are growing every year. These animals return to Polish forests after many years of risk of extinction or, as in the case of European bison, are reintroduced step by step. Unfortunately, they are still rare and require close health surveillance and monitoring. One serious threat to their health and life is the bacterial parasite *Anaplasma phagocytophilum*. Moose and bison can also be sources of pathogenic bacteria for humans, which can be transferred through tick bites. In line with the World Health Organization (WHO) “OneHealth” program, indicating that animal health affects human health and vice versa, the following study examines the occurrence of *Anaplasma phagocytophilum* bacteria in European bison and Eurasian moose populations using molecular biology tools. Our results provide useful data regarding *Anaplasma phagocytophilum* that could be used in future strategies for the diagnosis and treatment of people and animals.

**Abstract:**

Wild large ungulates, like European bison (*Bison bonasus*) and Eurasian moose (*Alces alces*), form an important part of the circulation of *Anaplasma phagocytophilum*, a Gram-negative, intracellular, tick-transmitted bacterium, in the natural environment. Bison and moose tissue samples were subjected to *16S* rDNA, *groEL* and *ankA* partial gene marker amplification with specific primers using various variants of PCR. Out of 42 examined individuals, *Anaplasma* sp. were detected in 4/13 Eurasian moose (31%) and 7/29 European bison (24%). In addition, 12 *groEL* and 5 *ankA* partial gene positive samples were obtained from the examined animals. The phylogenetic analysis of the *groEL* partial gene classified samples from European bison to ecotype I, and samples from Eurasian moose to ecotype I and II; the analysis of the *ankA* partial gene assigned the samples to clusters I and IV. This study extends knowledge about *A. phagocytophilum* in wild large ungulates in Poland. This is the first report about the occurrence of *Anaplasma* sp. in one of the largest populations of free living European bison in the world. Our findings confirm that strains of *A. phagocytophilum* from *Bison bonasus* and *Alces alces* may constitute a natural reservoir of pathogenic HGA *Anaplasma* strains.

## 1. Introduction

*Anaplasma phagocytophilum* is a Gram-negative, intracellular, tick-transmitted bacterium from the Anaplasmataceae family known as the causative agent of anaplasmosis [1,2]. The first report of *A. phagocytophilum* was published in the 1930s [3]. Since then, it has been known as *Cytoecetes phagocytophila*, *Ehrlichia phagocytophila* and *E. equi*. [2]. Finally, in the early 2000s, the species were unified into *A. phagocytophilum* [4].

The new classification of *Anaplasma phagocytophilum* in 2001 [4] was based on the genetic analysis of the *16S* rDNA gene, and since then this genetic marker, full or partial, has been used for detection and phylogenetic analysis [5,6,7,8,9]. In addition, the genes *groEL*, *ankA*, *msp2* and *msp4* are frequently used for identifying *A. phagocytophilum* strains [10,11,12]. *Anaplasma phagocytophilum* has been divided into four ecotypes circulating in the environment based on the *groEL* heat–shock operon gene [6,11,13]. Of these, ecotypes III and IV have clearly defined hosts, i.e., ecotype III is associated with rodents and VI with birds, while variants I and II have a range of varied hosts: ecotype I has been associated with goats (*Capra hircus*), hedgehogs (*Erinaceus europaeus*), roe deer (*Capreolus capreolus*) and humans, and ecotype II with ruminants [11,14]. All ecotypes occur in various vectors from the *Ixodes* genus [12,15]. Based on the analysis of the *ankA* gene, *A. phagocytophilum* strains can be divided into five clusters, with each strain being correlated with a host species [1,16,17,18]. Cluster I includes strains associated with humans, companion and farm animals, while cluster II and III are associated with roe deer and cluster IV with various ruminants, including sheep (*Ovis aries*), European bison (*Bison bonasus*) and cows (*Bos taurus taurus*). Cluster V is associated with rodents [1,14,16,17,18].

The Eurasian moose (*Alces alces*) (IUCN—least concern LC) and European bison (IUCN—near threatened NT) are under protected status in Poland (Dz.U. 2005 nr 48 poz. 459, Dz.U. 2004 nr 92 poz. 880). A ban on hunting moose has been in place since 2001 (suspension of hunting/moratorium), and the European bison is under strict protection. Both species were progressively reintroduced in Poland following WW I and WW II [19,20]. The strains of *Anaplasma phagocytophilum* identified in Europe are more virulent for wild and farm animals, especially cattle and ruminants, than humans [1,9]. However, more than 5000 human granulocytic anaplasmosis (HGA) cases were reported in 2019 in the USA (Centers for Disease Control and Prevention—CDC) compared to around 300 in Europe [21]. Although the European bison and Eurasian moose populations in Poland have stabilized relatively recently, both species remain rare and demand close supervision and health monitoring.

The aim of this study was to detect *Anaplasma phagocytophilum* in tissues from wild European bison and Eurasian moose, and to genotype and characterize the detected strains with the use of *16S* rDNA, *groEL* and *ankA* genetic markers. The conducted analysis will be used to provide a classification of *A. phagocytophilum* into ecotypes and clusters. This study is the first *groEL* and *ankA* marker-based genetic characterization of *A. phagocytophilum* strains in Eurasian moose in Poland.

## 2. Materials and Methods

### 2.1. Materials

The tissue samples from the moose were collected in the period 2018 to 2021 from two localizations in Poland: the Kampinos National Park (*n* = 5), Warsaw Urban Forests (*n* = 6) in Mazovian Voivodeship (D) and the Polesie National Park (*n* = 2) in the Lublin Voivodeship (B) (Figure 1). As hunting for moose is illegal in Poland, all samples were collected from road kill specimens or animals found dead by forest and national park employees. The samples from the European bison were collected in the years 2021 to 2022, also in two locations: the Bieszczady Mountains (Subcarpathian Voivodeship) (C) and the Białowieża Primeval Forest (Podlaskie Voivodeship) (A) (Figure 1). In the Białowieża Primeval Forest, the tissue samples were taken from specimens that were found dead (*n* = 5) or had been legally eliminated (*n* = 4) by employees of the Białowieża National Park. Any spleen or liver samples from European bison in the Bieszczady Mountains were collected from individuals (*n* = 20) culled according to the decision of the General Directorate for Environmental Protection (DZP-WG.6401.2.2021.EB). Spleen and liver samples were collected with the approval of the Regional Directorate for Environmental Protection in Rzeszów.

### 2.2. Molecular Methods

DNA was isolated from the spleen and liver using a commercial DNA Mini Kit (Syngen, Wroclaw, Poland) according to the manufacturer’s protocol. *Anaplasma* spp. was detected using semi-nested PCR to amplify the partial *16S* rDNA according to Szewczyk et al. (2019) [22]. Positive samples were additionally tested for the partial *groEL* and *ankA* genes with nested PCR according to Alberti el al. (2005) [10] and Massung et al. (2007) [23], respectively. DNA amplification was performed using the DNA Engine T100 Thermal Cycler (BioRad, Hercules, CA, USA). DNA isolated from wild boar (*Sus scrofa*) infected with *A. phagocytophilum* (MT510541.1) was used as a positive control [24]. The PCR products were visualized on a 1.2% agarose gel (Promega, Madison, WI, USA) stained with SimplySafe (EURx, Gdańsk, Poland) and a size-marked DNA Marker 100 bp LOAD DNA ladder (Syngen, Wroclaw, Poland). Visualization was performed using ChemiDoc, MP Lab software (Imagine, BioRad, Hercules, CA, USA). The obtained PCR products were purified with the DNA clean-up Kit (Syngen, Wroclaw, Poland). Products were sequenced by Genomed (Warsaw, Poland) and assembled using ContigExpress, Vector NTI Advance v.11.0 (Invitrogen Life Technologies, New York, NY, USA). The obtained sequences were compared with those from GenBank in BLAST (NCBI, Bethesda, MD, USA) and submitted to GenBank. Phylogenetic trees were constructed using Bayesian inference (BI), as implemented in the MrBayes version 3.2.0 software [25]. The HKY + G for *groEL* and GTR + I + G for *ankA* model was chosen as the best-fitting nucleotide substitution model using JModelTest version 2.1.10 software [26,27]. The analysis was run for 1,000,000 generations, with 250,000 generations discarded as burn-in. The phylogenetic trees were visualized using the TreeView software (S&N Genealogy Supplies, Chilmark, UK).

**Figure 1 animals-12-01222-f001:**
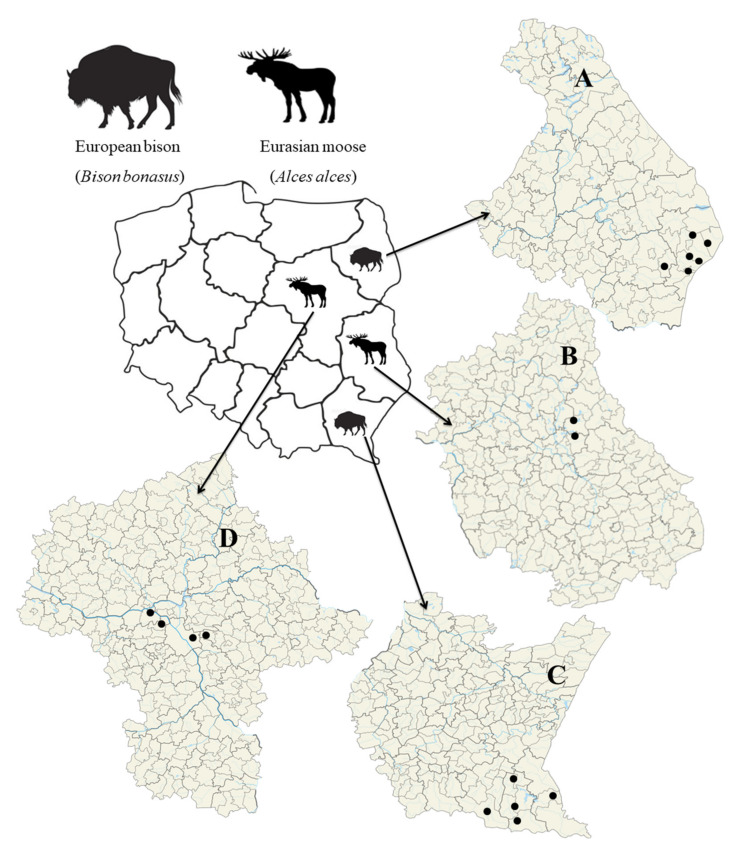
Sampling location in western and central parts of Poland. A—Podlaskie Voivodeship; B—Lublin Voivodeship; C—Subcarpathian Voivodeship; D—Masovian Voivodeship.

### 2.3. Statistical Methods

Categorical variables were presented as counts and percentages, and compared between groups using Fisher’s exact test. The 95% confidence intervals (CI 95%) for percentages were calculated using the Wilson score method [28]. A significance level (α) was set at 0.05, and all statistical tests were two-tailed. The statistical analysis was performed in TIBCO Statistica 13.3 (TIBCO Software Inc., Palo Alto, CA, USA).

## 3. Results

### 3.1. Prevalence

In total, samples collected from 42 animals, i.e., 13 moose and 29 European bison, were tested. Some of the findings regarding the detection of *Anaplasma* sp. in Eurasian moose have been published previously [9]. In the spleen samples, *Anaplasma* sp. *16S* rDNA partial gene was detected in four moose and six bison. In the liver samples, *16S* rDNA was detected in three moose and one bison. In total, 11 animals tested positive for *Anaplasma* spp. (26%; CI 95%: 15%–41%)—four of 13 Eurasian moose (31%; CI 95%: 13%–58%) and seven of 29 European bison (24%; CI 95%: 12%–42%) By region, the prevalence of Anaplasma spp. in the European bison population was 44% (CI 95%: 19%–73%; 4/9) in the Białowieża Primeval Forest, and only 15% (CI 95% 5%–36%; 3/20) in the Bieszczady Mountain (*p* = 0.158).

### 3.2. 16S rDNA

A random selection of samples found to be positive for *Anaplasma* spp. *16S* rDNA partial gene were sent for sequencing. The sequences from *Bison bonasus* (ON007214, ON025953) were identical to each other, and both demonstrated a 100% similarity with a common strain of *Anaplasma phagocytophilum*. Among the moose (*Alces alces*), one of the identified *16S* rDNA sequence variants was previously described in Myczka et al. (2021a) [9]. All sequences from European bison and Eurasian moose were found to be identical to each other; they were also identical to many submissions (Table 1) from Europe, Asia and North America associated with various hosts: carnivores, ungulates, humans, small mammals, insects and ticks.

### 3.3. groEL

In total, 12 *groEL* partial gene positive samples were obtained from the spleen and liver samples. Seven randomly chosen samples, i.e., three from Eurasian moose and four from European bison, were sequenced. The obtained sequences (OM835683–OM835689), together with GenBank data, were used to build a phylogenetic tree (Figure 2). The phylogenetic analysis placed most of the sequences into ecotype I. Only one sequence from *Alces alces* (OM835683) was classified to ecotype II, according to Jahfari et al. (2014) [11].

### 3.4. ankA

Of the 11 positive individuals, five partial *ankA* genes were obtained: two from bison and three from moose. This marker was only successfully amplified from spleen samples in European bison, but from both tissues from Eurasian moose. Only one moose demonstrated the *ankA* gene in both tissue samples. The obtained sequences have been deposited in the GenBank database under accession numbers: OM835678–OM835682. Our *ankA* partial gene nucleotide sequence was assigned to three groups (Figure 3).

### 3.5. Statistical Analysis

Our present findings do not indicate any statistically significant difference in the number of animals infected with *Anaplasma* spp., neither in European bison nor in Eurasian moose, with regard to sex or age, nor any difference with regard to the test used (detailed results not shown). Although clear differences in prevalence can be seen between the sample collection areas for the European bison, insufficient samples were taken from Białowieża Primeval Forest to determine whether the difference was statistically significant.

## 4. Discussion

Knowledge of the role and circulation of *Anaplasma phagocytophilum* in the natural environment continues to grow, with increasing numbers of studies being concerned not only with the detection and prevalence of *A. phagocytophilum* bacteria, but also with their phylogenetic relationships based on the use of genetic markers such as *ankA* and *groEL*. Our present findings extend current knowledge on the prevalence and genetic characteristics of *A. phagocytophilum* strains in Eurasian moose (*Alces alces*) and European bison (*Bison bonasus*): two large wild ruminant species currently under protection in Poland.

Out of 42 examined wild ruminants, 11 tested positive for *Anaplasma* sp., with a prevalence of 26%. While only a few studies have confirmed the presence of *Anaplasma phagocytophilum*, the causative agent of bovine anaplasmosis, in *Bison bonasus* in Europe, all samples were obtained from bison in the Białowieża Primeval Forest. Matsumoto et al. (2009) [29] found the prevalence to be 20%; however, the study only included five animals. Later, Scharf et al. (2011) [1] reported a prevalence of 57.5%, while Dzięgiel et al. (2015) [5] found this rate to be 18.34%, and Karbowiak et al. (2015) [30] found it to be 66.57%. In the present study, the overall prevalence of *A. phagocytophilum* in European bison was found to be 24%, which, despite being a little low, is nevertheless in line with previous reports. By region, the prevalence of *Anaplasma* spp. was found to be 44% (4/9) in the Białowieża Primeval Forest but only 15% (3/20) in the Bieszczady Mountain population, although the difference in prevalences was not statistically significant, likely due to the small number of animals enrolled in the study. In North America, bovine anaplasmosis, caused by *A. marginale*, can also occur in American bison (*Bison bison*), an animal closely related to the European bison. In one study, de la Fuente et al. (2003) [31] report the prevalence of *Anaplasma* spp. to be 88% and 100% among two American bison populations from the USA and Canada. Both rates are significantly higher than any reported in European bison [1,5,29,30], including our present findings. The present study is the first report about the occurrence of *Anaplasma phagocytophilum* in a European bison population from the Bieszczady Mountains, the second largest population of free-living European bison in the world [32,33]. In the Eurasian moose (*Alces alces*), our findings indicate a significantly lower prevalence (31%) than that reported in Norway (41.4–70%) [12,34,35] and Sweden (82%) [36] in Europe. In Poland, in only one report, the prevalence showed a lower rate (20%) [30]. In North America, moose are also subject to infection by *Anaplasma* spp. Its presence was confirmed in Eastern moose (*Alces alces americana*) with a prevalence of 54% [37], which is markedly higher than our present findings.

Our phylogenetic analysis of the partial *groEL* gene allowed the detected *Anaplasma phagocytophilum* strains to be assigned to two of the four ecotypes proposed by Jahfari et al. (2014) [11]. The obtained sequences of the *groEL* gene belong to five haplotypes. Of these, four haplotypes, viz. OM835683, OM835684, OM835686 and OM835687, had already been reported in the GenBank database, while another sequence, OM835689, is new; in addition, OM835687 and OM835688 are identical to each other. All samples of the *groEL* partial gene from European bison were found to be ecotype I. So far, no reports exist on the division of *A. phagocytophilum* strains obtained from European bison into ecotypes based on the *groEL* marker. Although a previous report from Slovakia did employ a partial *groEL* gene sequence, it was too short to assign the detected *A. phagocytophilum* strain to any particular ecotype [38]. However, some reports have described the amplification of a partial *groEL* gene in samples from cattle, the group of animals most closely related to *Bison bonasus*. A comparison of our samples of *groEL* (Figure 2) with representative sequences of *A. phagocytophilum* strains detected in cattle from Germany (KU587056, KU587060, GQ452230) and France (KJ832487, KJ832483) revealed that the identified *A. phagocytophilum* strains were ecotype I, like in the *Bison bonasus* samples [11,39,40]. The samples from *Alces alces* tested in the present study were classified to ecotypes I and II [11]. The presence of both ecotypes in the tested moose is in line with previous reports from Norway and Sweden [12,41]. Interestingly, both our samples of *Alces alces* from Poland (Mazovian Voivodeship) (Figure 1) and *Alces alces* from Norway and Sweden were collected in small areas, in which two different ecotypes of *A. phagocytophilum* strains were among a single species of animal [9,12,41].

The genetic marker *ankA* was able to divide *Anaplasma phagocytophilum* strains into clusters. Sequences from European bison (OM835681, OM835682) were placed within cluster IV [1] together with various bovine hosts (Figure 3). One of the *ankA* genes from moose (OM835679) was found to share cluster I with sequences from humans, dogs, red deer and horses, among others (Figure 3); this is in line with reports from Scharf et al. (2011) [1]. None of the detected *ankA* haplotypes were identical. Only one haplotype from *Bison bonasus* (OM835681) had a 100% match with sequences from the GenBank database (KJ832243, GU236731); the other sequences of the *ankA* partial gene (OM835678, OM835679, OM835680, OM835682) are new haplotypes not reported before. This is the first report of *ankA* gene sequences identified in Eurasian moose.

An important consideration in research about *Anaplasma phagocytophilum* in wild animals is whether the detected strains may be potentially pathogenic to humans. Wild and game animals can be potential reservoirs of pathogenic *A. phagocytophilum* strains [24,42,43]. The present study used three genetic markers for analysis, viz. *16S* rDNA, *groEL* and *ankA*, which are also suitable for identifying HGA *A. phagocytophilum* strains [1,43]. The *16S* rDNA partial gene described in European bison and Eurasian moose in the present study, and in Myczka et al. (2021a) [9], is 100% identical to *A. phagocytophilum* samples from humans in Europe (Austria, Belgium), North America (USA) and Asia (South Korea) (Table 1). In addition, most of the *groEL* genes detected in the present study were classified to ecotype I (Figure 2), which is also home to various HGA strains [11]. While the partial *ankA* gene from *Bison bonasus* was placed in cluster IV, which does not include any pathogenic *A. phagocytophilum* strains, the sequence from *Alces alces* was classified into cluster I, with the *ankA* gene from humans (Figure 3). Based on these findings and the phylogenetic analysis, it cannot be clearly confirmed whether the detected strains of *A. phagocytophilum* from bison and moose are potentially pathogenic for humans. However, the fact that the tested strains and their markers can be grouped with known HGA strains suggests that the animals tested in this study may constitute a natural reservoir of pathogenic *A. phagocytophilum*.

## 5. Conclusions

This study extends current knowledge about *Anaplasma phagocytophilum* in wild large ungulates in Poland. Our findings constitute the first report of the occurrence of *Anaplasma* spp. in free-living European bison from the Bieszczady Mountains, one of the largest populations in the world. They include the first genotyping of *A. phagocytophilum* strains from moose (*groEL*, *ankA*) in Poland and the first record of the *ankA* partial gene from Eurasian moose in the world. They also confirm that the detected strains from *Bison bonasus* and *Alces alces* may constitute a natural reservoir of strains of *A. phagocytophilum* pathogenic for humans.

## Figures and Tables

**Figure 2 animals-12-01222-f002:**
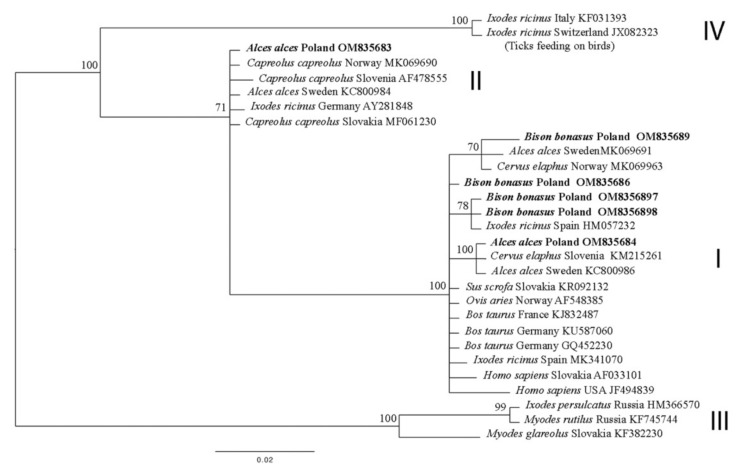
Phylogenetic tree of *groEL* partial gene (528 bp) haplotypes from *Anaplasma phagocytophilum*, constructed by Bayesian inference (BI) analysis using MrBayes version 3.2. The HKY + G model was chosen as the best-fitting nucleotide substitution model using JModelTest version 2.1.10 software. The scale bars are proportional to the number of substitutions per site. In bold, sequences from this study. I–IV ecotype of *A. phagocytophilum* according to Jahfari et al. (2014) [11].

**Figure 3 animals-12-01222-f003:**
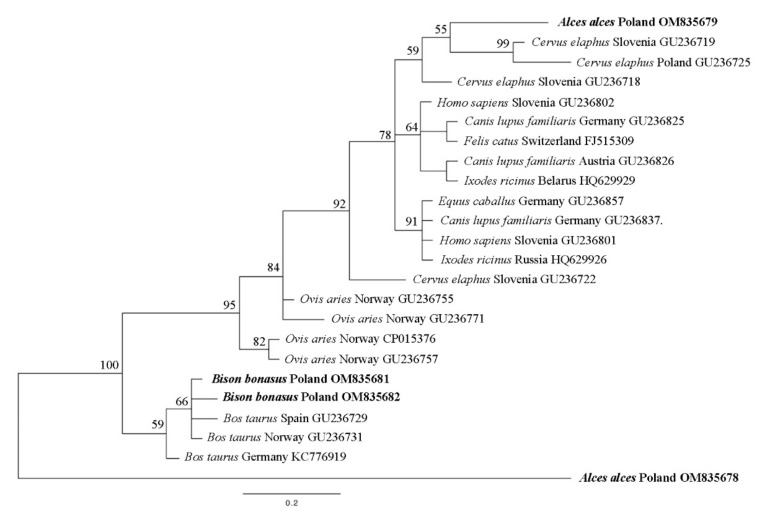
Phylogenetic tree of ankA partial gene (630 bp) haplotypes from Anaplasma phagocytophilum, constructed by Bayesian inference (BI) analysis using MrBayes version 3.2. The GTR + I + G model was chosen as the best-fitting nucleotide substitution model using JModelTest version 2.1.10 software. The scale bars are proportional to the number of substitutions per site. In bold, sequences from this study.

**Table 1 animals-12-01222-t001:** Examples of *Anaplasma phagocytophilum 16S* rDNA sequences with 100% similarity to our *16S* rDNA sequences, from the NCBI GenBank.

Hosts Group	Host	Country	Gen Bank No.
Carnivores	Red fox (*Vulpes vulpes*)	Switzerland	KX180948.1
		Poland	MH328211.1
	Dog (*Canis lupus familiaris*)	Croatia	KY114936.1
		Germany	JX173651.1
		Republic of South Africa (RSA)	MK814406.1
		Iraq	MN453475.1
		Japan	LC334014.1
	European badger (*Meles meles*)	Poland	MH328211
	Coyote (*Canis latrans*)	United States of America (USA)	AF170728.1
	Raccoon dog (*Nyctereutes procyonoides*)	South Korea	KY458557.1
	Cat (*Felis catus*)	South Korea	KR021165.1
	European polecat (*Mustela putorius*)	Poland	MH328208.1
Humans	*Homo sapiens*	Austria	KT454992.1
		Belgium	KM259921.1
		USA	AF093788.1
		South Korea	KP306520.1
Small mammals	Bank vole (*Clethrionomys glareolus*)	United Kingdom	AY082656.1
	Natal multimammate mouse (*Mastomys natalensis*)	RSA	MK814411.1
	Northern red-backed vole (*Myodes rutilus*)	Russia (Sverdlovsk region)	HQ630622.1
	Black-striped field mouse (*Apodemus agrarius*)	South Korea	KR611719.1
		China	GQ412337
			DQ342324
	European hedgehog (*Erinaceus europaeus*)	Germany	FN390878.1
Insect	*Ixodes ricinus*	Estonia	MW922755.1
		Belarus	HQ629915.1
		Austria	JX173652.1
		Turkey	FJ172530.1
	*Ixodes pacificus*	USA	KP276588.1
	*Ixodes persulcatus*	Russia (Irkutsk region)	HM366584.1
	*Ixodes tapirus*	Panama	MW677508.1
	*Haemaphysalis longicornis*	China	KF569908
		South Korea	GU064898
	*Haematopota pluvialis*	Poland	MH844585.1
Ungulates	Sheep (*Ovis aries*)	Norway	CP015376.1
	Red deer (*Cervus elaphus*)	Slovenia	KM215243.1
	Roe deer (*Capreolus capreolus*)	Spain	MN170723.1
	Wild boar (*Sus scrofa*)	Poland	MT510541.1
	Llama (*Lama glama*)	USA	AF241532.1
	Horse (*Equus ferus caballus*)	USA	AF172166.1
	Cow (*Bos taurus taurus*)	Turkey	KP745629.1
	Goat (*Capra hircus*)	China	KF569909.1

## Data Availability

The data that support the findings of this study are available from the corresponding author, upon reasonable request.

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
