# Peer review of "Prevalence and Genotyping of Anaplasma phagocytophilum Strains from Wild Animals, European Bison (Bison bonasus) and Eurasian Moose (Alces alces) in Poland"

_animals, 2022, doi:10.3390/ani12091222_

Round 1
Reviewer 1 Report
Dear authors, the ms is well written and the methods appropriate. Thr results well presented and discussion and conclusions are appropriate too. Please edit the style of scientific names that must be put in italics.
For that I only ask for minor revisions.
Best,
the reviewer

Author Response
Dear Reviewer,
Thank you very much for the fact that you all have done so much work and devoted
a lot of attention while reviewing our manuscript.
We send to you an improved manuscript, which has significantly increased in value and presents precise and reliable information about the research that we conducted.
Kind regards
Anna W. Myczka (Correspondent Author)

Reviewer 2 Report
The results of some previous investigations confirmed the infection of European bison and Eurasian moose, with Anaplasma phagocytophilum. However, in my opinion, the Authors of reviewed manuscript extend current knowledge about A. phagocytophilum in wild large ungulates. The methods used by the Authors are appropriate and well described. The Table and Figure are needed and help in better understanding of the text. Discussion is comprehensive and references are adequate. The title and abstract accurately convey what has been found. I found some errors:
133-173 lines: in the results section, you need to review the Latin words and write them in Italic: for example "Anaplasma", "Bison bonasus", "Alces alces" etc.
Author Response

(The authors gave the same response as above.)

Reviewer 3 Report
Dear authors,
Your article “Prevalence and genotyping of Anaplasma phagocytophilum 2 strains from wild animals, European bison (Bison bonasus) and 3 Eurasian moose (Alces alces) in Poland” is very interesting. There are some minor changes I recommend to be done.
L47: “Anaplasma phagocytophilum”: full name only the first time that it is refers in the text. All the other times should be “A. phagocytophilum”.
L125- 130: I have not seen any results from the statistic analysis. Please add a table. The result should be in “3. Results” not at the “4. Discussion”
L135-170: “Anaplasma sp” and “A. phagocytophilum” should be in italics
L 151: insert the “Table 1” after the paragraph
Figure 2: “Anaplasma phagocytophilum” and “A. phagocytophilum” should be in italics
Table 1: “A. phagocytophilum” it should be “Anaplasma phagocytophilum” when it is the first time it refers in the heading. “Table 1” should be removed at the end of “3.2. 16S rDNA”.
Figure 3: it should be removed in “3.4 ankA”. Please refer “Fig 3” in the text in “Results”
Author Response

(The authors gave the same response as above.)
